# EmoSpeech: Guiding FastSpeech2 Towards Emotional Text to Speech

*Daria Diatlova[1], Vitaly Shutov[2]*

[1]VK, deepvk, Saint Petersburg, Russia
[2]VK, deepvk, Saint Petersburg, Russia

`d.dyatlova@vk.team`, `vi.shutov@corp.vk.com`

## Abstract

State-of-the-art speech synthesis models try to get as close as possible to the human voice. Hence, modelling emotions is an essential part of Text-To-Speech (TTS) research. In our work, we selected FastSpeech2 as the starting point and proposed a series of modifications for synthesizing emotional speech. According to automatic and human evaluation, our model, Emo-Speech, surpasses existing models regarding both MOS score and emotion recognition accuracy in generated speech. We provided a detailed ablation study for every extension to Fast-Speech2 architecture that forms EmoSpeech. The uneven distribution of emotions in the text is crucial for better, synthesized speech and intonation perception. Our model includes a conditioning mechanism that effectively handles this issue by allowing emotions to contribute to each phone with varying intensity levels. The human assessment indicates that proposed modifications generate audio with higher MOS and emotional expressiveness.

**Index Terms**: text to speech, emotional text to speech, fast speech

## 1. Introduction

In recent years, the field of Text-to-Speech (TTS) has made significant progress in terms of the quality of synthesised speech thanks to models based on Normalising Flows [1, 2, 3], diffusion process [4, 5, 6], Transformer architecture [7, 8, 9] and more. While Transformer-based models, such as Fast-Speech2 [8], lose out to some of the latest diffusion models, such as NaturalSpeech2 [6], in terms of quality metrics, i.e. Mean Opinion Score (MOS), they outperform others in terms of inference speed. Our work mainly focuses on providing a solution for high-load environments, such as large social networks where inference speed matters, so we choose FastSpeech2 [8] architecture as a starting point.

We often consider how realistic generated speech sounds while estimating its quality. To produce realistic speech, TTS models must consider many factors absent from simple text input, such as prosody, stress, rhythm, intonation, and emotion. Emotional speech enriches the spoken text's meaning, making it easier to perceive the context. In this paper, we explore how to adopt FastSpeech2 for emotional speech synthesis (ETTS), raise the question of the uneven distribution of emotions across the text sequence, and analyze it later in this work.

The main contributions of this work are as follows:

- We propose the extension to the FastSpeech2 model architecture with several known and new modules that make it possible to synthesize speech with a desired emotion.

- The proposed model outperforms an existing implementation of FastSpeech2 extended for Emotional Speech Synthe-sis [1] [10], regarding MOS and emotion recognition accuracy, without bringing inference speed latency.

- We propose a conditioning mechanism that makes it possible to account for the relationship between speech intonation and the strength of the emotion that falls on each token in the sequence. EmoSpeech pays attention to each part of the sentence depending on the emotion.

## 2. Related Work

The question of transmission emotions to the synthesised speech has been studied since HMM-based TTS models, so [11] introduced the style control vectors for four predefined styles. Later in [10] presented the approach of injecting emotional embedding into attention RNN and the hidden state of the decoder RNN of Tacatron [12]. The emergence of Global Style Tokens (GST) [13] played a significant role in the development of approaches to conditioning text utterance on the speech reference [14]. GST aims to learn an embedding vector for each emotion from reference speech and text and introduce a global style token to obtain utterance-level embeddings. Unlike GST, where final embedding is a weighted sum of several tokens, in [15] extracted a single reference vector and proposed an interpolation algorithm to control the intensity of emotional expressiveness. Emotional intensity control is also addressed in [16, 17]. Some recent works [18, 19] have shown that a guided text description of the desired emotion can be conveyed to the speech.

To sum up, at a higher level, ETTS methods can be broadly classified into three distinct categories based on the nature of their conditioning data. These categories include models that use:

1. Categorical labels to represent one or more emotions;
2. Referenced speech with the desired emotional state;
3. Textual descriptions of the target emotional state as a form of conditioning data.

The first approach is traditionally used when working with a labelled dataset, as it helps implement conditioning simply by introducing an embedding lookup table. Since our main focus in this work is to provide a model for a fast ETTS capable of synthesizing speech in a high-load product environment when given a small fixed set of speakers and emotions, we take the first approach.

The traditional approach to synthesizing emotional speech given a categorical label is presented in [10]. Initially, it was built on top of Tacotron2 [12]. The authors of [10] suggested concatenating a categorical label with the pre-net output and

---

[1] `https://github.com/keonlee9420/Expressive-FastSpeech2`

then adding a layer to project the vector to match the size of the attention RNN input and the first layer of the RNN decoder of Tacotron2. Later, this approach was adapted for FastSpeech2 [8] in Expressive FastSpeech2 [20], where conditioning works similarly to multi-speaker conditioning through adding emotion embedding to the encoder output. Expressive FastSpeech2 implementation is the most relevant work for us, as it also uses FastSpeech2 and suits our inference speed constraints.

### 2.1. Technical details of adapted modifications

The choice of FastSpeech2 as a starting point for further model modification for expressive synthesis [20, 10], zero-shot speech synthesis case [21], better quality in multi-speaker scenario [22] etc., is not novel. One of the most successful modifications is AdaSpeech [23] for custom voice synthesis, and GANSpeech [22] for multi-speaker synthesis.

The central concept introduced in AdaSpeech is Conditional Layer Norm (CLN). CLN aims to extend well-known LayerNorm [24] but with a condition on speaker embedding for calculating scale and bias. Specifically, CLN takes the form of:

$$y = f(c) \cdot \frac{x - mean}{var} + f(c), \quad (1)$$

where f is a linear layer, x is a normalized hidden and c is a conditioning embedding. In AdaSpeech, all layer normalizations in the encoder were substituted with CLN. Later, in AdaSpeech4 [21], which was designed for zero-shot scenarios, it was suggested to integrate CLN in both the encoder and decoder for better performance.

Quality degradation is one problem that appears in FastSpeech2-like models after extensive conditioning. It is addressed in GANSpeech [22], which occurs while training FastSpeech2 in a multi-speaker setup. In [22], the authors aimed to improve speech quality using two training phases. During the first phase, FastSpeech2 is trained using a classic reconstruction loss. In the second phase, the JCU discriminator without a hierarchically-nested structure from VocGAN [25] is applied for adversarial training. JCU discriminators consist of conditional and unconditional parts. The unconditional part of the discriminator processes the mel spectrogram x as it is, and the second processes x within the conditioning vector c. FastSpeech2 (G) and the JCU discriminator (D) were then optimized by using the least squares objective:

$$L_{adv}(D) = \frac{1}{2}\mathbb{E}_c\left[D(\hat{x})^2 + D(\hat{x}, c)^2\right]$$
$$+ \frac{1}{2}\mathbb{E}_{(x,c)}\left[(D(x) - 1)^2 + (D(x, c) - 1)^2\right]$$
$$L_{adv}(G) = \frac{1}{2}\mathbb{E}_c\left[(D(\hat{x}) - 1)^2 + (D(\hat{x}, s) - 1)^2\right].$$

In [22], the authors also apply feature matching loss to improve model quality and stability by computing the L1 loss between JCU discriminator feature maps of real and generated mel spectrograms:

$$L_{fm}(G, D) = \mathbb{E}_{x,c}[\mathbb{E}_l(D_l(\hat{x}, c) - D_l(x, c))],$$

where $D_l$ is the output from the JCU discriminator layer $l$. GANSpeech is then trained with:

$$L_{total} = L_{rec} + L_{adv}(D) + L_{adv}(G) + \alpha_{fm} \cdot L_{fm} \quad (2)$$

## 3. Model Description

### 3.1. FastSpeech2

FastSpeech2 is a non-autoregressive acoustic model for fast and high-quality speech synthesis. The model takes a sequence of tokens as input and generates mel spectrograms, which are later upsampled to a waveform by a vocoder. The key components of FastSpeech2 are the encoder, variance adapter, and decoder [8]. The intuition behind each block is that the encoder extracts features from textual information about what would be said, the variance adaptor adds acoustic and duration information to the input sequence, and the decoder generates features from all of this for the mel spectrogram features. The encoder and decoder are a feed-forward transformer block, a stack of multi-head self-attention layers and 1D-convolution. An encoder converts a token embedding sequence into the token's hidden representation $h \in \mathbb{R}^{n \times hid}$, where $n, hid$ are the sequence length and hidden dimension, respectively. The variance adaptor consists of 3 predictors followed by a length regulator. Predictors take $h \in \mathbb{R}^{n \times hid}$ and output the pitch, energy, and durations $(p, e, d)$ for each token. Then, the length regulators "upsample" $h \in \mathbb{R}^{n \times hid}$, accumulated with $p, e \in \mathbb{R}^n$ according to $d \in \mathbb{R}^n$. Token duration is measured by the number of mel spectrogram frames, which is why length regulator output is $h \in \mathbb{R}^{m \times hid}$, where $m = \sum_{i=0}^n d_i$. After that, the upsampled hidden are passed through a decoder, and the hidden dimension is reflected in mel channels using a linear layer. The final output is a predicted mel spectrogram $y \in \mathbb{R}^{m \times c}$. The model learns to generate a mel spectrogram from the input text sequence using reconstruction loss:

$$L_{rec} = ||y - \hat{y}|| + ||d - \hat{d}||^2 + ||e - \hat{e}||^2 + ||p - \hat{p}||^2,$$

where $\hat{y}, \hat{d}, \hat{e}, \hat{p}$ are the predicted mel spectrogram, duration, pitch, and energy.

### 3.2. Conditioning Embedding

We use embedding lookup tables to construct EmoSpeech from FastSpeech2 for naive speaker and emotion conditioning. We form the conditioning vector c by concatenating the speaker and emotion embeddings. Concatenation allows us to build from 15 embeddings in the lookup table 50 unique embeddings for further conditioning. As a starting point of our modifications, we add a conditioning vector c to the encoder output before feeding it to the variance adaptor. Similarly to [20], we expand the embedding on a sequence length dimension.

### 3.3. eGeMAPS Predictor

By design, the variance adaptor of FastSpeech2 can be extended by adding additional predictors [8]. For EmoSpeech, we add the eGeMAPS predictor (EMP) to the variance adaptor that predicts $k$ features from the Extended Geneva Minimalistic Acoustic Parameter Set (eGeMAPS) [26]. In total, eGeMAPS consists of 88 features, however, in the search for a balance between a lightweight architecture and an increase in quality, we choose only two features from the eGeMAPS set for the prediction – 80th and 50th percentile of logarithmic F0. We describe the number of selected features in section 4. The idea behind EMP is to add to the utterance more information from the low-level speech descriptors that highly correlate with the target emotion. EMP has the same architecture as pitch and energy predictors – we follow the setup from [8]. However, unlike pitch and energy predictors, the eGeMAPS predictor operates on the utterance level rather than the token level.

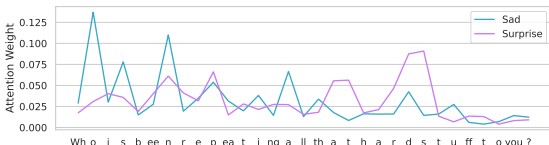 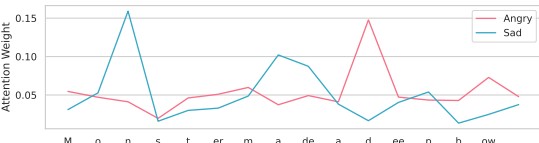

Figure 1: *A weights distributed over a single utterance in two emotions. Note that for the utterance in the first raw, the "surprise" emotion tends to have more weight at the end of the sentence. Conversely, the attention weights of the last tokens for "sad" emotion tend to have lower weights than in the beginning and middle of the utterance. See section 5 for more details.*

### 3.4. Conditional Layer Norm

Following the early success of AdaSpeech4 [21] in applying the Conditional Layer Norm (CLN) for zero-shot speech synthesis, we adopt CLN for emotional speech synthesis. We also found it beneficial to apply it instead of the traditional Layer Norm in the encoder and decoder blocks of EmoSpeech. CLN takes the same form as in Equation 1. For conditioning the embedding c,, we concatenate speaker and emotion embeddings that are taken from the embedding lookup tables.

### 3.5. Conditional Cross Attention

Expressive intonation is one of the characteristics of emotional speech. Moreover, if we listen to such speech, we notice that sometimes the speaker emphasises some parts of the sentence, making the emotion distinguishable. In contrast, the rest of the sentence may sound very neutral. At the same time, the traditional approach in emotional speech synthesis is to add emotion embeddings to each text token with the same weight [16, 10]. This work introduces a *Conditional Cross-Attention* (CCA) block to the encoder and decoder, which aims to reweight tokens according to the given emotion. Specifically, we add CCA to the encoder and decoder so that each transformer encoder and decoder block consists of Self-Attention, Conditional Cross-Attention and position-wise feed forward.

We concatenate speaker and emotion embedding for the utterance and give this conditioning vector a notation $c \in R^{hid}$ and notate Self-Attention output as $h \in R^{n \times hid}$. CCA utilize $W_q, W_k, W_v$ matrices from Self-Attention layer and forms: $Q = W_q \cdot h, K = W_k \cdot c, V = W_v \cdot c$. Then we reweight the hiddens:

$$w = \text{softmax}(\frac{Q \cdot K^T}{\sqrt{d}}, \dim = 1) \quad (3)$$

$$cca = w \cdot V$$

Roughly, this operation can be seen as adding a unique emotion token for every layer.

Similarly to the multi-head self-attention mechanism, we add multi-head logic to conditional cross-attention in our implementation. Note that CCA is a substitution for a naive addition of conditioning embedding c expand on a sequence length dimension to the encoder output before feeding it to the variance adaptor. This is why we no longer make this addition in EmoSpeech after the modification. Section 4 shows the density distribution of attention weights w across the text utterance for different emotions.

### 3.6. Adversarial Training

Although the above methods help us improve the transmission of emotionality and natural intonation (see Table 3), numerous artefacts can still be heard in the generated speech. The quality degradation might occur while generalizing FastSpeech2 for a

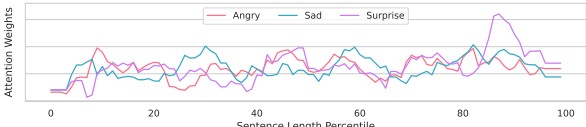

Figure 2: *Averaged attention weight distribution over the normalized sentence length in the test dataset for 10 speakers.*

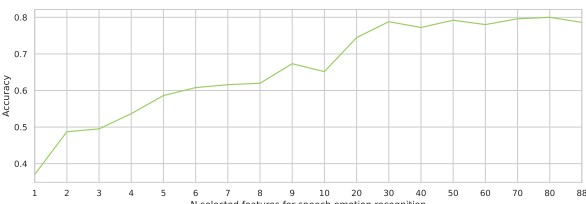

Figure 3: *The figure shows the accuracy of CatBoost Classifiers trained with N eGeMAPS features. Accuracy growth is slowing down with every new feature addition.*

multi-speaker setup, as was mentioned in GANSpeech [22]. To boost the quality of generated speech, we utilize the same technique as in [22] and apply adversarial training for EmoSpeech, as the conditional architecture of JCU discriminator [25] suits our multi-speaker and multi-emotional setup. We used the same architectural setup and training objective as in [22]. However, we trained the discriminator and the EmoSpeech during a single training phase. We use the same conditioning embedding c for the conditioning discriminator, which concatenates speaker and emotion embeddings.

Overall, EmoSpeech is trained with the same objective as in Equation 2, where $\alpha_{fm} = \frac{L_{rec}^{sg}}{L_{fm}}$ is added for training stabilization [22], $L_{rec}^{sg}$ notates stop gradient for $L_{rec}$. We also add an MSE for predicting eGeMAPS features to $L_{rec}$.

## 4. Data Preprocessing

**Dataset.** This paper focuses on providing a lightweight solution for speech synthesis with a fixed set of emotions and multiple speakers. For our experiments, we use the English subset of the Emotional Speech Database (ESD) [27], later referred to as a dataset. The dataset has a sufficient variety of speakers and lexical coverage. It contains 350 utterances from 10 speakers, each recorded in 5 emotions: Neutral, Angry, Happy, Sad, and Surprised, resulting in 1750 utterances for each speaker. It has a total word count of 11,015 words and 997 unique lexical words. Utterances have a diverse vocabulary as well as the tone of the sentences. Each utterance file has a text annotation and a corresponding single emotion label. We split the dataset into train-

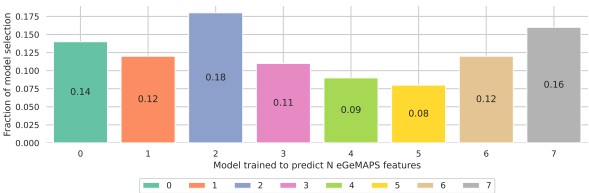

Figure 4: *Annotators' preference of speech synthesised by the model predicted from 0 to 7 eGeMAPS features. Note that the model trained to predict 2 features was chosen more often with a slight advantage.*

ing, validation and test according to the instructions in [27]. The validation and test sets consists of 20 and 30 utterances in 5 emotions and 10 speakers, resulting in 1000 and 1500 utterances. Because of the limited resources for MOS estimation, we used a subset of the test dataset containing 25 utterances. We sampled 5 out of 10 speakers, and for each speaker and emotion, we sampled a single utterance id, resulting in 14 unique utterances. So our test subset is emotion and speaker balanced, but not parallel; there are two reasons for it. First, if we used a single utterance for all speakers and emotions, we could not generalise the results into sentences with different vocabulary. Second, we gave annotators an utterance to estimate emotion transmission and asked them which emotion they recognised. If we used a single utterance across all emotions, we would encourage annotators to select using the exclusion method. We share the subset for further research and comparison of experiments.

**Feature extraction.** Since EmoSpeech training requires not only utterance along with the text transcript but also phonemes, mel spectrograms, durations, pitch, and other acoustic features, we apply the following preprocessing, extract:

1. phonemes, punctuation, and silence tokens from text annotations using the grapheme-to-phoneme (GTP) model from the Montreal-forced-aligner (MFA) [28] toolkit.
2. durations of extracted phonemes using MFA [28].
3. pitch from ground truth waveforms using `pyworld`[2] library.
4. energies normalizing spectrograms by frequency dimension.
5. eGeMAPS features using `openSMILE` toolkit [29].

Pitch, energies, and eGeMAPS features are normalized, and the first two are averaged frame-wise according to the phoneme durations. As for eGeMAPS features, we extract a single value for the utterance.

**eGeMAPS feature selection.** The eGeMAPS is a standard acoustic parameter set comprising 88 low-level feature descriptors and is used for various areas of automatic voice analysis. To find the most relevant features for emotion transmission, we extracted all features for the English subset of the ESD dataset and fitted CatBoost classifier [30] for the emotion classification task. Then, to find a balance between the quality and the number of features used for prediction, we used the Recursive Feature Elimination algorithm [31], see Figure 3. It can be seen that the accuracy of the multiclass classification of emotions increases as the number of features used increases. At the same time, the relative increase in accuracy decreases with each additional feature. Considering that our target task is emotional speech synthesis and not classification, as well as the unwillingness to

[2]https://github.com/JeremyCCHsu/Python-Wrapper-for-World-Vocoder

overload the Variance Adaptor block, we choose the 60% accuracy on multiclass classification as a threshold for further experiments. After that, we extracted seven features selected by the Recursive Feature Elimination algorithm for all utterances:

- 80th percentile of the log F0 .[3]
- 50th percentile of the log F0 .[4]
- The range of 20th to 80th percentile of the log F0 .[5]
- Spectral flux difference of the spectra of two consecutive frames .[6]
- Mel-Frequency Cepstral Coefficients 1 .[7]
- Harmonics-to-noise ratio (HNR), relation of energy in harmonic components to energy in noise-like components .[8]
- Equivalent sound level.[9]

All features were extracted using the openSMILE toolkit, so we listed library notation for all features in the footnote. Then we normalized all features and trained models with $n$ eGeMAPS features predicted where $n \in [1; 7]$. Afterwards, the synthesised speech of the baseline model (with 0 eGeMAPS features added) and each of the seven trained models were submitted for evaluation. Annotators were asked to select the recording they preferred most. As a result of the survey (see Figure 4), the model trained to predict 2 of the eGeMAPS was preferred by annotators more often.

## 5. Experiments and Results

**Model configuration.** The EmoSpeech architecture is based on FastSpeech2 [8] and is configured using the same hyperparameters, except phoneme embedding, encoder, and decoder hidden dimensions are set to 512; encoder and decoder Conv1D filter sizes are 512; encoder and decoder include 6 layers each. The hidden dimension of speaker and emotion embedding is 256. The eGeMAPS predictor follows the same architecture as the other predictors in the Variance Adaptor block: 2 1D convolutions with kernel size 3, stride 1, followed by ReLU activation and dropout 0.5. For JCU Discriminator, we follow the same architectural setup as in GANSpeech [22], including parameters for training configuration. We used iSTFTNet [32] vocoder trained on the English subset of the ESD database. For training configuration, we followed implementation[10] and parameters described in [32]. The mel spectrogram was extracted from a waveform with a filter length of 48 ms, a hop length of 12 ms, and 80 mel channels.

**Training and inference details.** We trained the model using 4 Nvidia A100 GPUs with a total batch size of 256 for 50000 steps; for the optimizer, scheduler, and related parameters, we follow the setup in FastSpeech2 [8]. The JCU Discriminator was trained together with EmoSpeech using Adam optimizer [33] with a learning rate of 0.0001 with $(\beta_1, \beta_2) = (0.5, 0.9)$.

**Baseline model.** As an implementation of Expressive Fast-Speech2 [10, 20] is based on FastSpeech2 and makes it possible to synthesize speech with a given emotion in very naive form by

[3]F0semitoneFrom27.5Hz_sma3nz_percentile80.0
[4]F0semitoneFrom27.5Hz_sma3nz_percentile50.0
[5]F0semitoneFrom27.5Hz_sma3nz_pctlrange0-2
[6]spectralFlux_sma3_amean
[7]mfcc1V_sma3nz_amean
[8]HNRdBACF_sma3nz_amean
[9]equivalentSoundLevel_dBp
[10]https://github.com/rishikksh20/iSTFTNet-pytorch

Table 1: *The description of the used notation for sequential model modifications according to section 3 along with total model size.*

| ID | Model | # parameters |
|----|-------|--------------|
| #1 | FastSpeech2 [8] + EMP | 47.1$M$ |
| #2 | #1 + CLN | 53.4$M$ |
| #3 | #2 + CCA | 53.4$M$ |
| #4 | #3 + JCU (EmoSpeech) | 53.4$M$ |

adding their embeddings to the encoder output before feeding it to the variance adaptor, we choose it as a baseline model.

**Evaluation setup.** We construct EmoSpeech starting from FastSpeech2 and modifying it by sequentially adding the components described in section 3. We assign unique IDs for each combination of modifications to simplify references, table 1 provides information about them and total model sizes. Although the size of EmoSpeech increased with the added modifications by 15% compared with the same configured Expressive FastSpeech2, we did not notice any notable decrease in the inference speed.

For all trained models, we conduct both automatic and human evaluations to collect detailed feedback with the help of annotators. For automatic evaluation we use

Table 2: *The MOS and NISQA [34] scores. Original stands for ground truth utterances, and reconstructed are utterances reconstructed from ground truth mel spectrograms by iSTFTNet.*

| ID | Model | MOS ($\uparrow$) | NISQA ($\uparrow$) |
|----|-------|------------------|---------------------|
| - | Original | 4.7 $\pm$ 0.49 | 4.17 $\pm$ 0.57 |
| - | Reconstructed | 4.54 $\pm$ 0.58 | 4.11 $\pm$ 0.58 |
| - | Expressive FastSpeech2 [20] | 3.74 $\pm$ 0.65 | 3.77 $\pm$ 0.74 |
| #1 | FastSpeech2 [8] + EMP | 4.06 $\pm$ 0.65 | 3.71 $\pm$ 0.76 |
| #2 | #1 + CLN | 4.25 $\pm$ 0.6 | 3.93 $\pm$ 0.66 |
| #3 | #2 + CCA | 4.33 $\pm$ 0.6 | 3.95 $\pm$ 0.66 |
| #4 | #3 + JCU (EmoSpeech) | **4.37 $\pm$ 0.62** | **4.1 $\pm$ 0.58** |

**Automatic evaluation.** For automatic evaluation, we used the `NISQA` library [11] [34]. Our work used the NISQA-TTS model and predicted naturalness score on a 5-point scale according to human MOS. The NISQA-TTS model was chosen as we focus on the realistic transmission of emotions that correlates with the naturalness of generated speech.

The final results on the test set are represented in Table 2. The EmoSpeech model outperforms the baseline and all intermediate modifications and shows similar quality to reconstructed utterances.

**Human evaluation.** For human evaluation, we test a subset of the ESD database, which construction is described in section 4, samples for evaluation can be found on the demo page [12]. We asked 20 annotators to solve three tasks.

The first task was to measure whether emotion was transmitted correctly. We asked the annotators to select the emotion they heard in the utterance: neutral or no emotion is recognized, anger, happiness, sadness, or surprise. If the annotator chose the target emotion for speech synthesis, we marked the response as 1, otherwise 0. We calculated an accuracy score as the average of all annotators' responses. The reported results are shown in Table 3.

---

[11] https://github.com/gabrielmittag/NISQA
[12] https://dariadiatlova.github.io/emospeech

Modification #3 has the highest accuracy across all emotions, with a slight quality increase from the EmoSpeech model. Accuracy drop after adversarial training is expected behavior as adversarial training was added to prevent synthesis quality degradation, its slightly smooth sharp intonation peaks in the generated mel spectrograms. It can also be seen that the baseline model has the highest accuracy score for the "anger" emotion. For that model, the annotators mentioned that artifacts that appeared in the generated utterance brought more anger to the sound, and therefore more samples were marked with the anger emotion.

Secondly, we evaluate utterance quality using the Mean Opinion Score (MOS). We asked annotators to evaluate sound quality on a 5-point scale, given additional instruction:

- 5 for excellent – no artifacts in the utterance;
- 4 for good – some artifacts are heard, but they do not influence utterance perception;
- 3 for fair – a lot of artifacts in the utterance;
- 2 for bad – many artifacts in the utterance, it is difficult to understand speech;
- 1 for poor – very noisy, almost impossible to listen to.

The results are shown in Table 2. Like the NISQA evaluation, the EmoSpeech model outperforms all intermediate modifications and baseline and shows synthesis quality close to the reconstructed utterances.

Lastly, we conduct a pairwise utterance comparison of the baseline model with each of our modifications to measure naturalness. We asked the annotators whether they would instead choose the first or the second utterance, given a target emotion. The results of the pairwise comparisons are shown in Figure 5. The EmoSpeech model is preferred by 10% more annotators than the baseline model.

**Analyzing attention weights in the CCA layer.** The intuition behind Conditional Cross Attention (CCA) is that different tokens of the text sequence have different effects on emotional intensity. We randomly picked two utterances from the test set to show emotion distribution over sentences and synthesised them in 2 opposite emotions. The result attention weights distribution for the utterances can be seen in Figure 1. Higher attention weight means higher token importance for target emotion expressiveness. For the emotion "surprise", which often correlates with interrogative intonation, the importance of tokens increases at the end of a sentence. Conversely, "sad" emotion tends to go down and has several spikes in the middle. The "angry" emotion has spikes at the beginning and in the middle of the sentence; it is also worth noting that the distribution of attention weights for "angry" emotion is very different from the distribution of attention weights for the same sentence in the sad emotion. To show that the trends described above are not just cherry-picked samples, in Figure 2, we show the attention weights distribution over normalized sentence lengths across the whole test dataset, which are speaker and emotion balanced, see section 4. As Figure 2 shows, intonation-specific peaks observed in Figure 1 can be generalized on the test set. The test set of the English subset of the ESD database includes interrogative and exclamatory utterances with diverse sentiments recorded in 5 emotions. This leads us to believe that the distribution of attention weights depends specifically on the acoustic features of emotions and not on the emotional content of the given text.

**Trust of validity.** Although EmoSpeech outperforms the baseline model in human and automated scoring, we would like to express our concerns about the metrics' validity. First of all,

Table 3: *Accuracy of emotions classified using human feedback. Overall, it is an average of all emotions.*

| ID | Model | Overall | Neutral | Angry | Happy | Sad | Surprise |
|----|-------|---------|---------|-------|-------|-----|----------|
|    | Original | 0.89 | 0.69 | 0.92 | 0.97 | 0.96 | 0.91 |
|    | Reconstructed | 0.86 | 0.59 | 1 | 0.92 | 0.94 | 0.87 |
|    | Expressive FastSpeech2 | 0.78 | 0.42 | **0.96** | 0.8 | 0.76 | 0.96 |
| #1 | FastSpeech2 + EMP | 0.78 | 0.45 | 0.94 | 0.68 | 0.81 | 1 |
| #2 | #1 + CLN | 0.82 | 0.48 | 0.92 | 0.88 | 0.82 | 1 |
| #3 | #2 + CCA | **0.85** | 0.55 | **0.96** | **0.91** | 0.83 | 1 |
| #4 | #3 + JCU (EmoSpeech) | 0.83 | **0.56** | 0.94 | 0.8 | **0.83** | 1 |

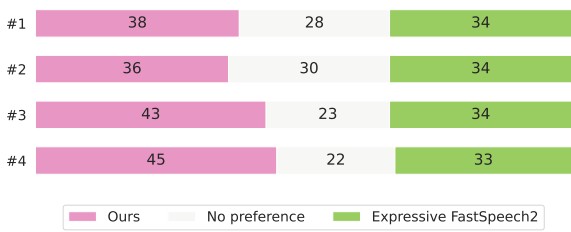

Figure 5: *Pairwise comparison of generated utterance naturalness given a target emotion. The numbers indicate the percentage of responses annotators choose this model against others. Note that adding modifications described in Table 3 increases preference.*

considering that the utterances for the test set are randomly selected and the set is balanced in terms of speakers and emotions, it is minimal, with 25 utterances in total. To increase confidence in the validity of the received MOS scores, we could increase the size of the test set. Second, to estimate the transfer of emotion to synthesised speech, we asked annotators to select the emotion from the fixed set of emotions that best matched the utterance. As the set of both utterances and annotators was fixed, there is a risk that during the evaluation of the original utterance, the annotators determined by the method of exclusion how each of the emotions sounds and used this knowledge in further evaluation of the synthesis. So, the results in Table 3, with an accuracy of 1 for "surprise" emotion, could not be interpreted as "surprise" emotion being best expressed by the models provided. We could interpret it as the most distinctive emotion from the given set.

## 6. Conclusion

In this paper, we developed EmoSpeech – an extension of the FastSpeech2 [8] model for Emotional Text-to-Speech Synthesis. We proposed multiple modifications for conditioning on a given emotion while keeping the model as fast as the original FastSpeech2. Experiments showed that all our modifications enhance model quality and outperform the previous extensions of FastSpeech2 for ETTS, Expressive FastSpeech2 [20]. One of the critical features of EmoSpeech is the conditional cross-attention mechanism, which considers the uneven distribution of emotion across a sentence. We demonstrated that for different emotions, the model selected different phonemes to accent, and therefore, our model sounds more natural than the baseline. Our source code and generated samples could be accessed via the `link`[13].

---

[13] https://github.com/deepvk/emospeech

## 7. Acknowledgement

We would like to thank Egor Spirin and Nikita Balagansky for reviewing the code and the paper bringing ideas that guided research into result direction; anonymous reviewers for valuable and helpfull review; Nikita Filippov for organizing human assessment; Victoria Loginova for writing and grammar correction.

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
