# OpenReview forum: "EmoSpeech: guiding FastSpeech2 towards Emotional Text to Speech"
_Interspeech.org/2023/Workshop/SSW — SSW12_

### Official Review · Reviewer_6yzY · 2023-05-30
**A simple extension of FastSpeech2 to generate emotional speech for 5 emotions (by predicting a couple of unspecified features, for an unknown language)**

**Rating:** 6
**Confidence:** 5

**Review:**

SUMMARY FOR THE AUTHOR AND THE META-REVIEWER

The core ideas are simple:

1) Extend FS2 to predict some additional acoustic features, beyond the F0 and energy of the original model. There are two additional features,chosen automatically from amongst the 88 eGeMAPS features, but which are NEVER ACTUALLY SPECIFIED. This severely reduces the value of this paper.

2) Add an attention-based mechanism to enable the model to learn how to weight the contribution of an emotion embedding (derived from an externally-provided label) to different places within an utterance. Some plots of this attention are provided, with a hand-wavy discussion, but no proper analysis.

There is sufficient novelty for publication at SSW.

The evaluation is rather poorly described. Specifically, there is no information on the text used, and whether this is the same across all emotions. The ESD database (which is not well described here; the reference doesn't even list the authors) supports this.

The results in Table 3 look too good to be true in some cases. There is perfect recognition of "surprise" which seems very unlikely. This is probably a consequence of a very small test set and the forced-choice task. Thus, listeners can identify an emotion 'by elimination' even if the speech doesn't sound particularly 'surprised'. 25 "audios" (note: audio is a mass noun and does not take the plural 's') = 5 speakers x 5 emotions. This is a very small test set.

Acceptance of this paper has to be conditional on fixing the absolutely disgraceful bibliography. See detailed comments below - it is the worst I have seen for some time.


DETAILED COMMENTS FOR THE AUTHOR

Global problems:

Egemaps -> eGeMAPS (e.g., section 3.3 heading)

Mel -> mel

ETTS is undefined

audio -> speech

audios -> utterances, since we are dealing only with speech here

incorrect opening double quotes in LaTex - your are all "99" but should be "66"

you never state which language you are using (ESD contains two)

The bibliography is, frankly, a disgrace. Missing conference/journal names. Many missing uppercase terms, including TTS. Even some missing author names. Please have more respect for the work you are building from, and the people who did it. Also, there is a worrying lack of older references. Emotional speech synthesis has been around for many, many decades, yet your citations are restricted to work in the last few years (5 at most). Please read the original work on this topic and relate it to your own.

Local problems:

Abstract - by phoneme you presuambly mean phone

Section 1

claiming most SoA models use flows is too strong - just a few do, there are plenty that don't

"Recently, research ..." -> untrue - see above comments about work many decades before yours

don't cite in 'shopping lists' like [8,9,10,11,12,13] - it is pointless and shows a lack of knowledge about what you are citing.

GST is not 'based on Tacotron' - it may have been first tried there, but has been applied to many other models since

footnote 1 is just code - please cite the paper too (not necessarily by the same author as the code)

[17] is cited as the first example of using categorical emotion labels - no, there is much, much prior work, a long time before this - go find it and read it

Section 3.1

generating a waveform from a spectrogram is not a matter of "upsampling"

what are "spectrogram bars" - do you mean frames?

shouldn't there be weights on the terms in the loss function?

Section 3.2

justify stacking (concatenating?) as opposed to summing embeddings

Figure 1

horizontal axis has no label or units

what does "follow an intuitive distribution" mean ?

Section 3.3.

fix heading as per global comment above

Section 3.5

FFT means Fast Fourier Transform in the speech field - spell out the acronym here to avoid confusion

Section 3.6

"help us improve" - where is the experimental result to support this claim?

Section 4.1

spell out ESD and fix the bibliography entry (no authors!)

do you really extract F0 (you say pitch) from the spectrogram, rather than a waveform? why?

eGeMAPS feature selection - you fail to specify what the "2 first features" actually are!

Section 4.3

"notable increase" -> "notable decrease"

what texts did you use in the evaluation? does the text have any implied emotion that could be a confound?

you report accuracy per emotion - specify precisely how this is computed, and whether there are any patterns in the confusion matrix

"models managed to add emotion" - this is surely not the intended behaviour! discuss!

"picks" -> "peaks" ?

how can you see "intonation peaks" in the spectrogram - explain and discuss

---

### Official Review · Reviewer_XQ5w · 2023-06-05
**emotion TTS using conditional layer norm, cross attention and adversarial training**

**Rating:** 6
**Confidence:** 4

**Review:**

This paper proposes extensions of FastSpeech 2 to enable emotion conditioned TTS.  There are 3 main components to this: a
* Conditional Layer Norm (cf. AdaSpeech, but adding an emotion embedding to the conditioning vector)
* Conditional Cross-attention (deriving the attention Keys using the emotion embedding)
* Adversarial training (JCU, cf. GanSpeech)

Overall, this paper should be of interest to the SSW community and its contributions are worth seeing at the workshop.

### Key Strengths
* The extensions made in the paper are clear, as is their relationship to previous work.
* Results training and testing on ESD are good: Table 3 indicates that recognizability of the emotions from synthesized speech is very good and the additions (CLN, CCA, JCU) improve on the baseline, with CLN+CCA appearing to perform the best overall .
* Experiments also show improvement in MOS
* CCA weights could be used to understanding how different parts of an utterance contribute to expression of emotion

### Main Weaknesses
* The proposed extensions are not super-novel in the sense that they are mostly adding the emotion embeddings to existing methods for conditioning the model (e.g. speaker embeddings).
* A lot section 2 really just gives the technical details for what is presented as new methods in section 3, meaning there isn't actually much discussion overall of related work on emotion synthesis.


### Novelty/Originality
As above, though I would note that I haven't seen these extensions implemented and tested otherwise, so it's definitely and interesting contribution.

### Technical Correctness
The work appears technically solid and reproducible.  It depends heavily on other papers, but the extensions are clear.

### Suggestions for improvement
* The exploration of the attention mechanism is an interesting start, but the figure looking at aggregated attention weight per emotion are not that easy to interpret.  It would be easier to see the patterns if they were grouped by emotion rather than speaker.  It would also be more instruction restrict the analysis to examples with the same text but different emotion and give word boundaries.  This would allow clearer understanding of how the attention mechanism deals with the lexical contribution of the utterance (vs acoustics).
* eGeMAPS features are added as variance adaptors (similar to F0, energy). The 88 features are reduced to 2 with feature selection but the paper never says what they actually are.  Also, no ablation is performed on this addition - is it even necessary?

### Quality of References
References are ok, though as noted above the discussion of emotion TTS seems a bit light.

### Clarity of Presentation
The paper is generally clear and the argumentation makes sense.

---

### Decision · Program_Chairs · 2023-06-14

**Decision:**

Accept

**Comment:**

SSW2003 received 45 papers. The acceptance rate is 82%. We are pleased to inform you that your paper has been accepted by the SSW2023 Program Committee. Please read the reviews carefully and submit your camera-ready paper by June 28th. Most reviewers performed a detailed review. Please answer to their questions and consider their comments. Note that camera-ready papers are credited with one extra page to allow authors to consider reviewers’ suggestions. So max 7 pages in total including figures & refs.
The deadline for submitting the revised version (with full non-anonymized authors and refs!) is 28th June.